# Extracellular Vesicles and Their miRNA Content in Amniotic and Tracheal Fluids of Fetuses with Severe Congenital Diaphragmatic Hernia Undergoing Fetal Intervention

**DOI:** 10.3390/cells10061493

**Published:** 2021-06-14

**Authors:** Isabella Fabietti, Tiago Nardi, Chiara Favero, Laura Dioni, Laura Cantone, Laura Pergoli, Mirjam Hoxha, Eva Pinatel, Fabio Mosca, Valentina Bollati, Nicola Persico

**Affiliations:** 1Fetal Medicine and Surgery Service, Fondazione IRCCS Ca’ Granda, Ospedale Maggiore Policlinico, 20122 Milan, Italy; isafabietti@gmail.com; 2EPIGET LAB, Department of Clinical Sciences and Community Health, University of Milan, 20122 Milan, Italy; tiago.nardi01@universitadipavia.it (T.N.); chiara.favero@unimi.it (C.F.); laura.dioni@unimi.it (L.D.); laura.cantone@unimi.it (L.C.); laura.pergoli@unimi.it (L.P.); mirjam.hoxha@unimi.it (M.H.); fabio.mosca@unimi.it (F.M.); 3Institute for Biomedical Technologies (ITB), National Research Council (CNR), Segrate, 20054 Milan, Italy; eva.pinatel@itb.cnr.it; 4Neonatal Intensive Care Unit, Fondazione IRCCS Ca’ Granda, Ospedale Maggiore Policlinico, 20122 Milan, Italy

**Keywords:** extracellular vesicles, miRNA, fetal endoscopic tracheal occlusion (FETO)

## Abstract

Infants with congenital diaphragmatic hernia (CDH) are at high risk of postnatal mortality due to lung hypoplasia and arterial pulmonary hypertension. In severe cases, prenatal intervention by fetal endoscopic tracheal occlusion (FETO) can improve survival by accelerating lung growth. However, postnatal mortality remains in the range of about 50% despite fetal treatment, and there is currently no clear explanation for this different clinical response to FETO. We evaluated the concentration of extracellular vesicles (EVs) and associated microRNA expression in amniotic and tracheal fluids of fetuses with CDH undergoing FETO, and we examined the association between molecular findings and postnatal survival. We observed a higher count of EVs in the amniotic fluid of non-survivors and in the tracheal fluid sampled in utero at the time of reversal of tracheal occlusion, suggesting a pro-inflammatory lung reactivity that is already established in utero and that could be associated with a worse postnatal clinical course. In addition, we observed differential regulation of four EV-enclosed miRNAs (miR-379-5p, miR-889-3p; miR-223-3p; miR-503-5p) in relation to postnatal survival, with target genes possibly involved in altered lung development. Future research should investigate molecular therapeutic agents targeting differentially regulated miRNAs to normalize their expression and potentially improve clinical outcomes.

## 1. Introduction

Congenital diaphragmatic hernia (CDH), which occurs in about 1 in 4000 pregnancies, is associated with significant postnatal morbidity and mortality [1]. Prenatal imaging and genetic testing provide the ability to stratify fetuses with CDH into different severity groups and to predict postnatal mortality, which can be as low as 10–15% in the extreme forms of the disease [2]. The most common causes of death in these cases are severe pulmonary hypoplasia and hypertension. Prenatal intervention by fetoscopic tracheal occlusion (FETO) has the objective to increase lung size by mechanical stretch due to the accumulation of fluid normally produced by the fetal lungs. A large multicentre study showed that, in severe cases, in utero treatment can increase postnatal survival from 24% to 49% for left-sided defects and from 0% to 35% for right-sided CDH [3]. The effects of FETO on survival are thought to be mediated by the increase in lung volume and the consequent improvement in gas exchange after birth. However, a significant proportion of neonates with severe CDH have some degree of pulmonary arterial hypertension despite a good lung growth after FETO and, in about 50% of these cases, death occurs in the neonatal period due to right cardiac failure secondary to pulmonary hypertension unresponsive to maximal respiratory support and medical treatment [4].

MicroRNA (miRNA) are small (18–25 nucleotides) non-coding RNAs that play an important role in expression regulation by targeting mRNAs for degradation or translation repression [5]. miRNA have been implicated in various processes in pulmonary tissues, such as development, homeostasis, and inflammation, and several studies have reported differentially regulated miRNAs in various physiological and pathological processes occurring in the lungs [6]. miRNAs exert their function both at the cellular and extracellular level. Extracellular miRNAs can be protein-bounded, freely circulating, or enclosed in extracellular vesicles (EVs). The incorporation of miRNAs in EVs allows those miRNAs to avoid degradation while circulating in biological fluids [7]. Previous studies showed that EVs play a key role in cell-to-cell cross-talk by carrying important information in the form of proteins, mRNA, and miRNA, which can regulate gene expression in the target cells [8,9,10].

The objective of this study was to evaluate EV concentration and associated miRNA expression in amniotic and tracheal fluids of fetuses with CDH undergoing fetal intervention and to examine the association between molecular findings and postnatal survival.

## 2. Materials and Methods

### 2.1. Study Population

Between 2014 and 2016, we recruited 15 pregnancies complicated by CDH, classified as severe based on the combination of the Observed/Expected fetal Lung area-to-Head circumference Ratio (O/E LHR), measured by obstetric ultrasound, and the side of the defect, which were both independent predictors of postnatal survival [2]. In all cases, an amniocentesis was performed, showing a normal fetal karyotype as an array-CGH was performed. The option of the fetal intervention was discussed with the parents if the O/E LHR was <25% in left-sided defects and <45% in right hernias, in the absence of associated major structural defects and/or chromosomal or genetic abnormalities, which were known to have a significant impact on postnatal survival. FETO was performed using a semi-rigid 2.0 mm diameter fetoscope (Karl Storz GmbH, Tuttlingen, Germany) through a 3.3 mm diameter cannula (Cook Medical, Bloomington, IN, USA), which was introduced transabdominal into the amniotic sac under ultrasound guidance after administration of fetal intramuscular anesthesia and maternal local or spinal analgesia. The fetoscope was advanced through the fetal oral cavity to reach the trachea, where a balloon (GOLDBALL 2 and COAX catheter, Balt Extrusion, Montmorency, France) was inflated with saline solution and detached from the catheter between the carina and the vocal cords (Figure 1). Balloon removal was planned about 6 weeks later using the same fetoscopic access to the trachea and, immediately after balloon puncture, 2 mL of tracheal fluid were aspirated through the fetoscope and stored at −80 °C. About 20 mL of amniotic fluid were collected at balloon insertion and stored at −80 °C. For each case, the fetal intervention was carried out after informed parental consent and formal approval by the Hospital Ethics Committee (“Comitato Etico—Milano Area 2” of the Fondazione IRCCS Ca’ Granda Ospedale Maggiore Policlinico, 20122 Milan, Italy) and the Italian Ministry of Health for off-label use of the medical devices. During the study period, FETO procedures were carried out as the standard of care and were given at the sole discretion of the physicians involved in the patients’ care (i.e., fetal surgeons, neonatologists, and pediatric surgeons). Patients provided informed consent to sampling and storage of amniotic and tracheal fluids, and they were made aware that subsequent analysis of such samples would not alter standard clinical management. Clinical data were retrospectively obtained from routinely collected medical records and fully anonymized before performing further analyses. 

### 2.2. EV Enrichment and RNA Extraction

EV preparation and analyses were carried out according to the “Minimal Information for Studies of Extracellular Vesicles (“MISEV”) guidelines” [12] (see Appendix A for details on compliance between the methods used in the manuscript and MISEV guidelines).

Samples (amniotic and tracheal fluids) were centrifuged 3 times at increasing speeds (1000× *g*, 2000× *g*, 3000× *g*) for 15 min at 4 °C to remove cell debris and aggregates. Supernatants were ultracentrifuged at 110,000× *g* for 75 min at 4 °C. The supernatant was removed, and the EVs were stored at −20 °C. EV characterization was performed on fresh samples. Enriched miRNAs were isolated from the frozen EV pellet with the miRNeasy purification kit (Qiagen) following the manufacturer’s instructions. For normalization of sample to sample variation, ath-mir-159a was added as a spike during RNA extraction. An Agilent 2100 BioAnalyzer (Agilent Technologies, Santa Clara, CA, USA) with Agilent RNA 6000 Pico Kit was used to assess the RNA integrity from the RNA Integrity Number.

### 2.3. EV Characterization

The number and dimension of EVs were assessed by nanoparticle tracking analysis (NTA). This technique measured the Brownian motion of vesicles suspended in a fluid and displays them in real-time through a charge-coupled device camera with high sensitivity. The Nanosight LM10-HS system (NanoSight Ltd., Amesbury, UK), was used to visualize the EVs by laser light scattering. Five 30 s recordings were performed for each sample. Collected data were analyzed with NTA software, which provided high-resolution particle size distribution profiles and EV concentration measurements. EVs were characterized by MACSQuant analyzer flow cytometer (Miltenyi Biotec, Bergisch Gladbach, Germany). Fluoresbrite^®^ Carboxylate Size Range Kit I (0.2, 0.5, 0.75, and 1 μm), was used to set the calibration gate on MACSQuant analyzer. To evaluate EV integrity, 60 μL sample aliquots were stained with 0.02 μM 5(6)-carboxyfluorescein diacetate N-succinimidyl ester (CFSE) at 37 °C for 20 min in the dark. Each aliquot of CFSE stained sample was then incubated with a specific antibody: CD14-APC (Clone TÜK4), CD105-APC (clone: 43A4E1), CD66abce-FITC (clone: TET2), CD326 (EpCAM)-APC (clone HEA-125), HLA-g-APC (clone MEM-G/9), and Herv-W-APC (clone 4F10). All of them were purchased from Miltenyi Biotec. Quantitative multiparameter analysis of flow cytometry data was carried out using FlowJo Software (Tree Star, Inc., Ashland, OR, USA).

### 2.4. MicroRNA Profiling

QuantStudio 12K Flex OpenArray Real-Time PCR System (Thermo Fisher Scientific, Waltham, MA, USA) was used to assess EV-associated miRNA profiling. Reverse transcription (RT) was performed using Megaplex RT primers Pool A v2.1 and Megaplex RT primers Pool B v3.0 (Life Technologies, Carlsbad, CA, USA) with stem-loop RT primers. The subsequent pre-amplification was carried out using Low Sample Input (Life Technologies) protocol with Megaplex PreAmp primers and TaqMan Preamp kit (Applied Biosystems, Waltham, MA, USA). Real-time PCR was performed using Open Array AccuFillTM System (Life Technologies), with OpenArray Human MicroRNA Panel. A total of 754 human miRNAs were amplified in each sample together with 16 replicates each of 4 internal controls (ath-miR159a, RNU48, RNU44, and U6 rRNA). Every amplification curve produced an AmpScore value, a quality measurement indicating the low signal in the amplification curve linear phase (range: 0–2). MiRNAs with a Crt value > 28 or AmpScore < 1.24 or missing were considered unamplified, and the Crt value was set to 29. MiRNAs that were not amplified in all subjects (*n* = 412 for amniotic fluid and *n* = 408 for tracheal fluid were excluded). NormFinder algorithm [13] was applied to choose the best normalization strategy among global mean, ath-miR159a, RNU48, RNU44, and RNU6. The global mean was selected as the best normalization method for miRNA measured in the amniotic and tracheal fluid. miRNA expression was determined using the relative quantification 2-ΔΔCrt [14].

### 2.5. MicroRNA Validation

MiRNAs resulting as differentially expressed in the miRNA profiling were candidates for the validation. Validation analysis was carried out with QuantStudioTM 3D Digital PCR System with TaqMan primers (Life Technologies). Reverse transcription (RT) was performed using Megaplex RT primers Pool A v2.1 and Megaplex RT primers Pool B v3.0 (Life Technologies) with stem-loop RT primers. The amplification reaction was performed using the following cycle conditions: 96 °C for 10 min, 39 cycles 60 or 62 °C 2 min (depending on the miRNA) and 98 °C for 30 s, 60 °C for 2 min.

### 2.6. Statistical Analysis

Descriptive statistics were performed on all variables. Categorical data were presented as frequencies and percentages. Continuous variables were expressed as median (Q1; Q3). Characteristics of study participants grouped by survival status at 4–6 years of age of life were compared using the Pearson chi-square test or Fisher exact test for categorical variables and Wilcoxon test for continuous variables.

Poisson linear regression analyses were applied to evaluate the association between EV count measured in amniotic and tracheal fluid (total EV, exosomes, microvesicles, CD66^+^, CD14^+^, CD105^+^, EpCAM^+^, HERV-w^+^, and HLAG^+^) and survival group (dead vs. alive). All models allowing for over-dispersion by estimating an additional dispersion parameter with Pearson chi-square. We reported means with 95% CI and *p*-values. For each EV size, we estimated EV geometric mean concentrations in the survival and non-survival group with Poisson linear regression models allowing for over-dispersion. Due to the high number of comparisons, we used a multiple comparison method based on Benjamini–Hochberg False Discovery Rate (FDR) to calculate the FDR P-value. The criterion applied to identify significant sizes was an FDR *p*-value < 0.20 and a *p*-value < 0.05. To display the results of the analyses, we used a series graph for EV mean concentrations of each group and vertical bar charts to represent FDR and *p*-value. For all graphs, x-axis was the size of EVs.

The geometric mean RQ value for each miRNA (measured in amniotic fluid or tracheal fluid) was calculated separately for survivors and not-survivors and their ratio was used to obtain the Fold Change (FC). For each miRNA, we executed a *t*-test to assess the mean miRNA difference between the 2 groups. MiRNA expression values were log2 transformed to achieve a normal distribution. In the screening phase, a miRNA was considered to be differentially expressed if the *p*-value was <0.05 and FC was <0.5 or >2. In the validation phase, we applied negative binomial regression models, and we estimated geometric mean and relative 95% CI in survivors and non-survivors. miRNAs with *p*-value < 0.05 were considered differentially expressed.

Statistical analyses were performed with SAS 9.4 software (SAS Institute Inc., Cary, NC, USA).

### 2.7. Prediction of Target Genes

The miRWalk2.0 database was interrogated to obtain a customized list of putative miRNA targets combining the predictions of DIANA-microTv4.0, miRanda (rel2010), Pictar2, PITA, RNA22v2, and Targetscan6.2 [15,16]. The potential target genes were required to be predicted by at least two prediction algorithms. To Identify miRNA-predicted targets involved in CDH, we crossed our list with that published by Russell et al. [17], containing a manually curated list of genes involved in this pathology in mouse models. Murine genes were converted into human homologs by Homologene [18].

## 3. Results

### 3.1. Clinical Data

In the study population of 15 fetuses with severe CDH undergoing FETO, the defect was left-sided in 10 (66.7%), right-sided in 4 (26.7%), and bilateral in 1 (6.6%). Median maternal age was 33 years (IQR 30–35) and the median gestational ages at balloon insertion and removal were 28.9 (IQR 27.9–29.9) and 34.1 (IQR 33.1–35.0) weeks, respectively. The median gestational age at delivery was 36.6 (IQR 33.3–38.3) weeks, and the median birth weight was 2500 (IQR 2000–3075) grams. Postnatal surgical repair was carried out in 14 (93.3%) cases at a median age of 3 (IQR 2–3) days, and the size of the diaphragmatic defect was such that the placement of a prosthetic patch was required in 12 (85.7%) of the 14 operated cases. Overall postnatal survival was 46.7% (7/15). The median postnatal age at death was 23 (IQR 11–259) days. In all non-survivors, the cause of death was cardiorespiratory failure secondary to severe pulmonary hypertension. Table 1 shows the comparison of obstetric and fetal factors between infants who survived and those who died. 

There was no significant difference between the two groups in gestational age at fetal intervention, pre-FETO O/E LHR, side of the defect, and duration of tracheal occlusion. In contrast, survivors showed a higher increase in O/E LHR during tracheal occlusion (35.8% vs. 16.2%; *p* = 0.02), were born at a higher gestational age (38.3 vs. 33.3 weeks; *p* = 0.01), and they weighed more at birth compared to non-survivors (3015 vs. 2160 g; *p* = 0.02). An amniotic fluid sample at balloon insertion was available in 6 (85.7%) of 7 survivors and 6 (75.0%) of the 8 non-survivors. The respective number of tracheal fluid samples at balloon removal were 6 (85.7%) and 5 (62.5%).

### 3.2. Size Distribution and Count of Extracellular Vesicles in FETO Survivors and Non-Survivors

To quantify the EV content of amniotic and tracheal fluids of fetuses undergoing FETO procedure, we performed a nanoparticle tracking analysis (NTA). 

First, we compared the count of exosomes (here defined as vesicles with a size of 30–130 nM), microvesicles (130–700 nm), and total EV means between survivors and non-survivors (Table 2 and Appendix A). 

In this latter group, we found an increase in all the counts, which was significant for all comparisons except for exosome counts in amniotic fluid (*p* = 0.096). The highest difference was observed for microvesicles in the tracheal fluid, as they were 3.17 times higher in non-survivors compared to survivors (*p* < 0.0001).

As we are aware that the classification of exosomes/microvesicles by size is often referred to as imprecise since a size overlapping between the two groups might exist, we further compared the two groups in terms of the distribution of mean vesicle concentrations for each size. In amniotic fluids, EV concentration was higher in non-survivors but the difference was not statistically significant (*p*-value < 0.05 for larger EVs, FDR *p*-value > 0.2; Figure 2).

In tracheal fluid, there was a highly significant difference between the two groups in EV concentration, with a peak at the size of 230 nm (*p*-value < 0.05 and FDR *p*-value > 0.2 for EVs between 190 and 430 nm; Figure 3).

### 3.3. Characterization of Extracellular Vesicles by Flow Cytometry

We performed flow cytometry on EVs, using membrane markers characterizing cell types potentially involved in the process (Table 2 and Appendix A). In amniotic fluids of non-survivors, we observed a higher concentration of EVs derived from neutrophils (Fold-change 1.41, *p*-value < 0.0001), monocytes (Fold-change 1.42, *p*-value < 0.0001), epithelium (Fold-change 1.55, *p*-value < 0.0001), placenta (HLA-G^+^ EVs: Fold-change 1.43, *p*-value < 0.0001; HERV-W^+^ EVs: Fold-change 1.13, *p*-value = 0.0008). Moreover, tracheal fluids of non-survivors showed a higher concentration of EVs derived from monocytes (Fold-change 1.33, *p*-value = 0.0005), endothelium (Fold-change 1.50, *p*-value < 0.0001), and epithelium (Fold-change 1.32, *p*-value = 0.0010).

### 3.4. miRNA Expression Screening in Amniotic and Tracheal Fluids

The expression profiles of 754 miRNAs were examined by Openarray, in the FETO survivors versus non-survivors. Openarray data showed that 342 miRNAs were detectable (expressed in at least one sample) in amniotic fluids and 346 miRNAs in tracheal fluids (Appendix A). In the amniotic fluid, we identified 3 miRNAs that showed a significantly higher expression (i.e., mir-379-5p, mir-190-5p, and mir-889-3p) in non-survivors. In the tracheal fluid, 8 differential miRNAs (mir-548d-5p, mir-503-5p, mir-223-3p, mir-29b-3p, mir-200a-5p) showed a higher expression in non-survivors while mir-17-3p, mir-200b-5p and mir-505-5p were upregulated in survivors (Table 3).

### 3.5. miRNA Validation in Amniotic and Tracheal Fluids

Differential miRNAs resulting from the screening (Table 3 and Appendix A) were chosen for validation by Digital PCR. The validation step included 8 survivors and 4 non-survivors: 3 subjects were excluded due to the low miRNA yield. In the amniotic fluid, 2 out of 3 miRNAs were significantly up-regulated in non-survivors and showed differential expression between the two groups (miR-889-3p and miR-379-5p; Table 4, in bold). In the tracheal fluid, 2 out of 8 miRNAs were significantly up-regulated in non-survivors and showed differential expression between the two groups (miR-223-3p and miR-503-5p; Table 4, in bold). 

### 3.6. Prediction of Target Genes

MiRWalk v.2.0 was used to easily collect the targets of miR-889-3p, miR-379-5p, miR-223-3p, and miR-503-5p predicted by six prediction algorithms (see Materials and Methods). The genes predicted by at least two algorithms were regarded as bona fide targets obtaining from 3300 to 4850 predicted targets per miRNA. To better define the role of the miRNAs in CDH we looked for bona fide targets present in the list of CDH related genes manually curated by Russell et al. [17] and, in Table 5, we show for each of our miRNAs the predicted CDH related targets.

## 4. Discussion

The main findings of this study were firstly that EV concentration in the amniotic and tracheal fluids of CDH fetuses undergoing fetal intervention was higher in cases that died postnatally compared to survivors. Secondly, we observed differential regulation of four EV-enclosed miRNAs in the amniotic and tracheal fluids concerning postnatal survival, with target genes possibly involved in lung development and clinical severity of CDH. 

The EV count was higher in non-survivors for most EV sizes but more markedly and significantly in the range of microvesicles. A previous study showed, in vitro on human cells and in vivo on mice, an increased intrapulmonary release of endothelial-derived microvesicles in response to a chemical or mechanical lung injury [19]. In addition, Lee et al. showed that epithelial cell-derived microvesicles promote macrophage-regulated lung inflammatory response [20]. Similarly, Moon et al. observed that epithelial EVs trigger alveolar macrophage activation and pro-inflammatory cytokine releases, confirming their roles in mediating cell-cell cross-talk [21]. In our study, flow cytometry analysis showed that the most represented EVs in the tracheal fluid derived from monocytes, endothelium, and epithelium. An increased release of EVs from these cell types may represent an excessive inflammatory response to mechanical lung stress caused by tracheal occlusion, and, possibly, this may be associated with a poor response to postnatal treatment and with a worse clinical outcome. If our results are confirmed on a larger cohort of CDH neonates, EV count in the tracheal fluid could provide clinicians with the ability to identify, prenatally or immediately after birth, the subgroup of CDH neonates that may not respond to standard treatment protocols. This knowledge can stimulate research into alternative therapeutic strategies for these cases, including different ventilation protocols, use of anti-inflammatory medications, immediate use of extra-corporeal membrane oxygenation, and others.

Analysis of EV-enclosed miRNA expression about postnatal survival showed an over-expression of two miRNAs in the amniotic fluid and two in the fetal tracheal fluid of non-survivors. In the pre-FETO amniotic fluid samples, there was an over-expression of mir-379-5p and mir-889-3p. mir-379-5p reduces proliferation, invasion, and migration of vascular smooth muscle cells and induces their apoptosis, targeting insulin-like growth factor 1 (IGF1). The upregulation of miR-379-5p found in non-survivors could, therefore, produce a decrease in the expression of IGF1. IGF-1 regulates Endothelin-1 (ET-1) [22], which has been found to have an important role in vascular hypertrophy and proliferation, leading to hypertension [23,24]. miR-889-3p was shown to target fibroblast growth factor receptor 2 (FGFR2). Fibroblast growth factors (FGFs) are critical in the regulation of placental implantation, including trophoblast differentiation and migration [25]. Additionally, FGF2 signaling pathway is a key factor in the regulation of placental endothelial cell proliferation and angiogenesis [26,27]. In the tracheal fluid collected in-utero at the time of reversal of tracheal occlusion, we observed an increased expression, in non-survivors, of mir-223-3p and mir-503-5p, which have also been shown to be involved in regulating pulmonary smooth muscle cell proliferation and migration [28,29]. A previous study on a rat model of pulmonary hypertension reported a down-regulation of miR-223-3p in the experimental group compared to the controls, showing that an increased expression was associated with reduced proliferation of pulmonary arterial smooth muscle cells and improved cardiovascular parameters [30]. It is possible that in fetuses with CDH, excessive suppression of cell proliferation and migration during fetal life may lead to a more severe lung under-development. Alternatively, overexpression of miR-223-3p may be a positive feedback response to more aberrant cell proliferation in the fetal pulmonary arterial vessels. However, future studies are needed to investigate the relationship between miR-223-3p expression and vascular smooth muscle cell proliferation in fetuses with CDH.

Only one previous study examined miRNA expression in amniotic and tracheal fluids of fetuses with CDH undergoing FETO, showing an increased expression of miR-200 family in fetuses who died compared to survivors. In addition, the authors showed in vitro that miR-200 had an inhibitory effect on TGF-β signaling in human bronchial epithelial cells, suggesting a possible role of miR-200 on lung development via regulation of TGF-β production [31]. In the present study, we also observed an increased expression of EV-enclosed miR-200a, together with a decreased expression of miR-200b, in the tracheal fluid of survivors, but these differences lost statistical significance in the validation model, possibly due to the small number of samples available. Further research on differential expression of miRNAs in relation to the clinical outcome of neonates with CDH may lead to the future identification of molecular targets for potential new therapeutic agents.

This is, to the best of our knowledge, the first study, which investigated the possible role of EVs in human fetuses with CDH. However, the study sample was small due to the rarity of the condition and the need to analyze EVs in fresh samples to maximize the quality of collected data. The sample size also conditioned the validation process, as the technical validation was conducted on a subset of samples (those with enough material). Therefore, our results should be verified on a larger cohort of neonates with CDH.

## Figures and Tables

**Figure 1 cells-10-01493-f001:**
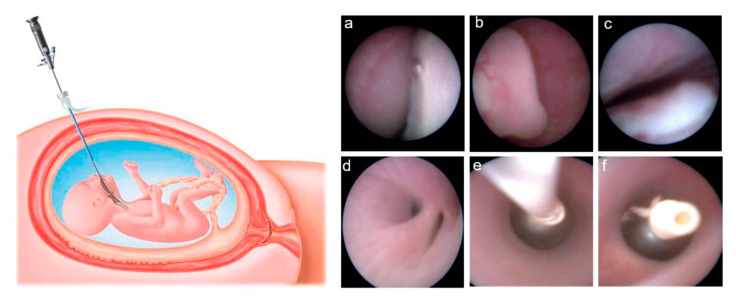
On the left side of the figure, schematic drawing showing intrauterine access to the fetal trachea by maternal percutaneous insertion of a fetoscope (reproduced from: Deprest, J.; Gratacos, E.; Nicolaides, K.H. Fetoscopic tracheal occlusion (FETO) for severe congenital diaphragmatic hernia: Evolution of a technique and preliminary results. *Ultrasound Obstet. Gynecol.* **2004**, *24*, 121–126 [11]). On the right side, endoscopic images showing the fetal oral cavity (**a**), the epiglottis (**b**), and the vocal cords (**c**), through which the fetoscope is advanced to reach the carina above the main bronchial bifurcation (**d**). At this level, the balloon is released through a catheter (**e**), inflated and detached to obtain tracheal occlusion (**f**).

**Figure 2 cells-10-01493-f002:**
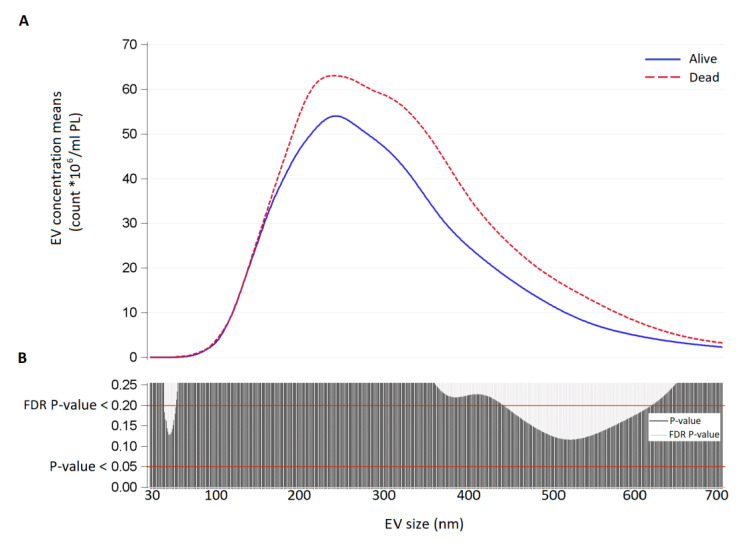
(**A**) EV concentration means (count × 10^6^/mL) for each size (30–700 nm) in amniotic fluid samples of survivors (continuous blue line) and non-survivors (dashed red line), collected at balloon insertion. (**B**) Significance of the difference between survivors and non-survivors for each EV size expressed as *p*-value and False Discovery Rate P-value from Poisson regression models allowing for over-dispersion.

**Figure 3 cells-10-01493-f003:**
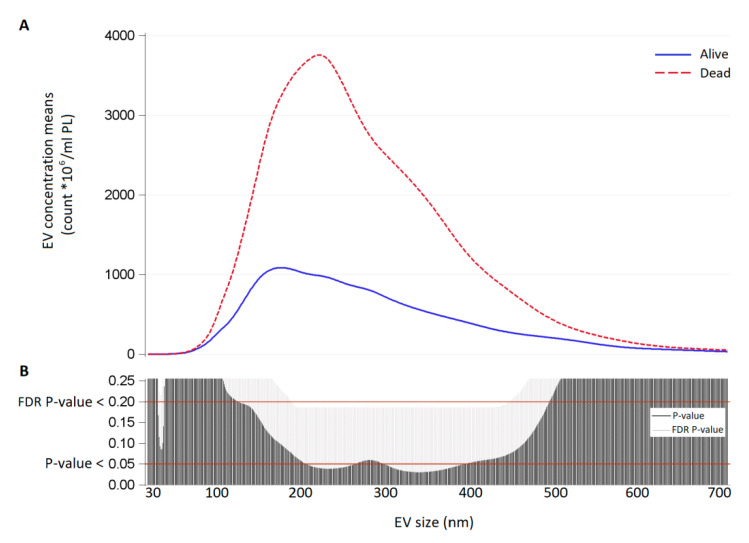
(**A**) EV concentration means (count × 10^6^/mL) for each size (30–700 nm) in tracheal fluid samples of survivors (continuous blue line) and non-survivors (dashed red line). (**B**) Significance of the difference between survivors and non-survivors for each EV size expressed as *p*-value and False Discovery Rate *p*-value from Poisson regression models allowing for over-dispersion.

**Table 1 cells-10-01493-t001:** Obstetric and fetal characteristics in fetuses that survived compared to those who died.

	Survivors *n* = 7	Non-Survivors *n* = 8	*p*-Value
**Obstetric Factors**			
GA at FETO, weeks median (q1–q3)	29.3 (28–29.9)	28.0 (27.7–29.2)	0.2699
Duration of TO *, days median (q1–q3)	42 (37–44)	37 (35–42.5)	0.4154
GA at delivery, weeks median (q1–q3)	38.3 (36.6–39)	33.3 (32.5–36.5)	0.0127
**Fetal Factors**			
Birth weight, g median (q1–q3)	3015 (2500–3440)	2160 (1740–2480)	0.0205
Female gender, *n* (%)	4 (57.1)	5 (62.5)	0.8327
Side of the defect			
Left	4 (57.1)	6 (75.0)	0.4126 *
Right	3 (42.8)	1 (12.5)	
Bilateral	0 (0)	1 (12.5)	
O/E LHR pre-FETO, % median (q1–q3)	33 (22.6–40.4)	25.3 (21.2–28)	0.1476
O/E LHR pre-removal, % median (q1–q3)	67 (59.3–76)	39.3 (37–50.8)	0.0015
Difference between pre-balloon removal and pre-FETO O/E LHR, % median (q1–q3)	35.8 (26.4–44.4)	16.2 (13.8–24.4)	0.0177

Quantitative data are presented as median (q1–q3), categorical variables are presented as frequencies and percentages. Q1 = first quartile; Q3 = third quartile; TO = tracheal occlusion; FETO = Fetoscopic Endoluminal Tracheal Occlusion; PROM = Premature Rupture of Membranes; O/E LHR = Observed/Expected Lung-to-Head Ratio. * Fisher exact test.

**Table 2 cells-10-01493-t002:** Ratio of the mean EV count between survivors and not-survivors subjects. *p*-value is estimated by linear Poisson regression models, allowing for overdispersion.

	Amniotic Fluid		Tracheal Fluid	
	Ratio Non-Survivor/Survivor	*p*-Value *	Ratio Non-Survivor/Survivor	*p*-Value *
**A—Nanoparticle Tracking Analysis**				
Total EV	1.28	<0.0001	3.12	<0.0001
Exosomes (<130 nm)	1.06	0.0965	1.94	<0.0001
Microvesicles (>130 nm)	1.29	<0.0001	3.17	<0.0001
**B—Flow Cytometry**				
CD66^+^ (neutrophils)	1.41	<0.0001	0.97	0.6873
CD14^+^ (monocytes/macrophages)	1.42	<0.0001	1.33	0.0005
CD105^+^ (endothelium)	n/a	n/a	1.5	<0.0001
EpCAM^+^ (epithelium)	1.55	<0.0001	1.32	0.0010
HLAG^+^ (placenta)	1.43	<0.0001	n/a	n/a
HERV-W^+^ (placenta)	1.13	0.0008	n/a	n/a

* *p*-value of the model comparing mean count in survivors versus mean count in non-survivors. Means and confidence intervals are reported in Appendix A.

**Table 3 cells-10-01493-t003:** Mean concentrations of EV-miRNA levels in survivors and non-survivors measured by OpenArray (miRNA selected for validation only).

	Amniotic Fluids		Tracheal Fluids	
miRNA	Fold Change	*p*-Value *	Fold Change	*p*-Value *
**mir-200a-5p**	0.450	0.2569	2.087	0.0005
**mir-17-3p**	Not expressed	Not expressed	0.268	0.0084
**mir-200b-5p**	0.247	0.2630	0.004	0.0104
**mir-548d-5p**	1.926	0.1540	3.381	0.0148
**mir-503-5p**	1.883	0.1116	2.129	0.0154
**mir-505-5p**	1.287	0.9787	0.191	0.0169
**mir-223-3p**	1.314	0.3905	3.506	0.0226
**mir-29b-3p**	1.261	0.5726	2.685	0.0241
**mir-379-5p**	2.425	0.0378	0.614	0.3493
**mir-190-5p**	2.546	0.0478	1.728	0.1021
**mir-889-3p**	2.094	0.0571	0.659	0.5155

* *p*-value of the model comparing mean count in not-survivors versus mean count in survivors. Complete miRNA expression levels of the screening are reported in Appendix A.

**Table 4 cells-10-01493-t004:** miRNA validation in Amniotic and Tracheal Fluids.

	miRNA	Ratio Non-Survivor/Survivor	*p*-Value *
**Amniotic Fluids**	mir-889-3p	2.02	**0.03**
	mir-379-5p	2.32	**0.03**
	mir-190-5p	1.75	0.10
**Tracheal Fluids**	mir-223-3p	2.58	**0.03**
	mir-503-5p	1.64	**0.05**
	mir-548d-5p	1.63	0.13
	mir-29b-3p	1.46	0.26
	mir-17-3p	1.72	0.3
	mir-200a-5p	1.25	0.5
	mir-505-5p	0.80	0.55
	mir-200b-5p	1.15	0.73

* *p*-value of the model comparing mean count in survivors versus mean count in not-survivors.

**Table 5 cells-10-01493-t005:** List of the miRNA predicted target genes.

miRNA	Target
mir-223-3p	Fibroblast growth factor receptor 2, GLI family zinc finger3, Paired box protein 7, Protein-lisine 6-ossidase, Low-density lipoprotein-related protein 2, Nuclear receptor subfamily 2 group 2 member 2/Transcription factor 2 COUP, Interleukin enhancer binding factor 3, Matrix metallopeptidase 14, Platelet-derived growth factor receptor A, Transcription factor 21
mir-503-5p	Low-density lipoprotein-related protein 2, Fibrillin 1, SRY-Box 7, Ephrin B1, Glypican-3, H 2.0 Like Homeobox, Homeobox protein Hox-B4, Willms tumor 1, Musculin, Platelet-derived growth factor receptor A, Transcription factor 21, Slit Guidance Ligand 3, Collagen type III Alpha 1 Chain
mir-379-5p	Deoxyribonuclease 2, Fibrillin 1, MET Proto-Oncogene receptor tyrosine kinase, Willms tumor 1, Retinoic acid receptor beta, Transcription Factor 21, GLI family zinc finger 3, Platelet-derived growth factor receptor A
mir-889-3p	Retinoic acid receptor beta, Transcription Factor 21, C-Terminal Binding Protein 2, Fibroblast growth factor receptor 2, Zinc Finger Protein FOG family member 2, GRB2-associated binding protein, Nuclear receptor subfamily 2 group 2 member 2/Transcription factor 2 COUP, SRY-Box 7, Collagen type III Alpha 1 Chain, Lysyl Oxidase, LDL Receptor Related Protein 2, Paired box protein 3, GLI family zinc finger 3, Platelet-derived growth factor receptor A, Fibrillin 1, Slit Guidance Ligand 3

## Data Availability

Data available on request due to restrictions. The data presented in this study are available on request from the corresponding author. The data are not publicly available due to privacy restrictions (they can be only presented in aggregated form).

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
