# Peer review of "Extracellular Vesicles and Their miRNA Content in Amniotic and Tracheal Fluids of Fetuses with Severe Congenital Diaphragmatic Hernia Undergoing Fetal Intervention"

_cells, 2021, doi:10.3390/cells10061493_

Round 1

Reviewer 1 Report

The manuscript is welly written and nicely put together several interesting considerations for a better understanding on CDH molecular basis.

In this regard, we know that apparently isolated CDH may be a prenatal sign of several genetic disorders (Pallister Killian Syndrome, Cornelia de Lange Syndrome and other chromatin related disorders etc), and, as it has been already suggested that constitutional higher levels of some miRNA (due to constitutional genetic abnormalities) may drive development of CDH (eg. Increased levels of miR200c have been hypothesized to suppress the expression of ZFPM2 gene in Pallister Killian S. fetuses, leading to development of CDH), it would be nice if you could specify which genetic examinations have been performed to rule out an underlyng constitutional genetic condition in enrolled fetuses.

Also, just some typing errors across the text to be edited. Other than this the manuscript looks great to me.

Author Response

>The manuscript is welly written and nicely put together several interesting considerations for a better understanding on CDH molecular basis.

We would like to first thank the editor and reviewers for their response to our work and their constructive comments.  We have addressed the minor concerns that were raised through further editing of the manuscript. 

>In this regard, we know that apparently isolated CDH may be a prenatal sign of several genetic disorders (Pallister Killian Syndrome, Cornelia de Lange Syndrome and other chromatin related disorders etc), and, as it has been already suggested that constitutional higher levels of some miRNA (due to constitutional genetic abnormalities) may drive development of CDH (eg. Increased levels of miR200c have been hypothesized to suppress the expression of ZFPM2 gene in Pallister Killian S. fetuses, leading to development of CDH), it would be nice if you could specify which genetic examinations have been performed to rule out an underlyng constitutional genetic condition in enrolled fetuses.

We fully agree with this point and for this reason, fetuses carrying genetic disorders have been excluded from the study. We added this information to the method section: “In all cases, an amniocentesis was performed, showing a normal fetal karyotype as an array-CGH was performed.”

>Also, just some typing errors across the text to be edited.

As suggested by the reviewer, we have gone through the revised article and corrected minor typing mistakes and grammatical errors.

>Other than this the manuscript looks great to me.

Thank you.

Reviewer 2 Report

The study by Fabietti et al evaluated EV concentration and associated miRNA 59
expression in amniotic and tracheal fluids of fetuses with CDH undergoing fetal intervention and examined the association between molecular findings and postnatal survival. The manuscript is scientifically sound and clearly written. The results are most likely to have clinical implications.

Their findings answer the study objectives. However, authors need to follow the guidelines mentioned in the link, https://www.tandfonline.com/doi/full/10.1080/20013078.2018.1535750, to characterize the EVs. 

Table 3 title needs to be modified to be suitable for the content of the table.

Author Response

>The study by Fabietti et al evaluated EV concentration and associated miRNA
expression in amniotic and tracheal fluids of fetuses with CDH undergoing fetal intervention and examined the association between molecular findings and postnatal survival. The manuscript is scientifically sound and clearly written. The results are most likely to have clinical implications.

We would like to first thank the editor and reviewers for their response to our work and their constructive comments. We have addressed the minor concerns that were raised through further editing of the manuscript.

>Their findings answer the study objectives. However, authors need to follow the guidelines mentioned in the link, https://www.tandfonline.com/doi/full/10.1080/20013078.2018.1535750, to characterize the EVs. 

We appreciate this comment. The method used in the paper follows the MISEV recommendation. Therefore, we added this reference to the methods, and also included a table describing how each technical aspect has been approached and if it is in line with MISEV guidelines.

>Table 3 title needs to be modified to be suitable for the content of the table.

As suggested by the reviewer, we corrected table 3 title.